# Comparing P300 flashing paradigms in online typing with language models

**Nand Chandravadia[1], Shrita Pendekanti[2], Dustin Roberts[2], Robert Tran[2],
Saarang Panchavati[2], Corey Arnold[2], Nader Pouratian[3], William Speier[2]***

**1** Deparment of Computer Science, Columbia University, New York, NY, United States of America,
**2** Department of Radiological Sciences, University of California, Los Angeles, Los Angeles, CA, United States
of America, **3** Department of Neurological Surgery, University of Texas, Southwestern, Dallas, TX, United
States of America

* Speier@ucla.edu

pone.0303390

University, UNITED STATES

**Data Availability Statement:** The data are available
from OpenNeuro at doi:10.18112/openneuro.
ds005028.v1.0.0 (accession number ds005028;

## Abstract

The P300 Speller is a brain-computer interface system that allows victims of motor neuron
diseases to regain the ability to communicate by typing characters into a computer by
thought. Since the system has a relatively slow typing speed, different stimulus presentation
paradigms have been proposed designed to allow users to input information faster by reducing the number of required stimuli or increase signal fidelity. This study compares the typing
speeds of the Row-Column, Checkerboard, and Combinatorial Paradigms to examine how
their performance compares in online and offline settings. When the different flashing patterns were tested in conjunction with other established optimization techniques such as
language models and dynamic stopping, they did not make a significant impact on P300
speller performance. This result could indicate that further performance improvements on
the system lie beyond optimizing flashing patterns.

## Introduction

Victims of amyotrophic lateral sclerosis (ALS), brain-stem stroke, and other upper motor neuron diseases lack the ability to vocalize their thoughts and emotion. With a sustained loss of
speech, their capacity to write, speak, and laugh is irreversibly impacted. However, the advent
of augmentative and alternative communication devices (AAC), such as brain-computer interfaces (BCI), have provided a possible avenue to restore their ability to communicate with the
external world.

The P300 speller, an electroencephalogram (EEG)-based BCI, translates neural signals
recorded from the scalp into speech in the form of virtual commands on a computer screen
[1]. This system utilizes the P300 signal, an endogenous event-related potential (ERP) with a
characteristic positive potential after a 300 millisecond delay from stimulus presentation [2].
First introduced by Farwell and Donchin, this system has users attend to a 6x6 matrix composed of alphanumeric characters. The user attends to a character on the matrix while the
rows and columns of the matrix flash randomly. Because the target character flashes relatively
infrequently in a stream of non-target, repeated stimuli, attending to the target character on

https://www.medrxiv.org/content/10.1101/2022.
06.24.22276882v1).

**Funding:** This material is supported in part by the National Science Foundation Graduate Research Fellowship Program (DGE-2034835).

**Competing interests:** The authors have declared that no competing interests exist.

the matrix elicits the P300 signal, according the "oddball" paradigm. The P300 signal, observed in the EEG, is then used in classification to detect which character on the matrix was selected. Though the P300 signal is robust, these systems generally have a relatively slow typing speed. Therefore many studies have focused on system optimization, attempting to improve overall system speed.

System optimization studies have traditionally focused on enhancing specific components of the P300 speller apparatus. For instance, Allison et al. modified the matrix size, demonstrating that increasing the size of matrix improves the amplitude of the P300 signal [3]. Lu et al. evaluated the inter-stimulus-interval (ISI), suggesting a longer ISI translates to a both a higher online accuracy and higher selection rate [4]. Both Townsend et al. and Jin et al. developed novel flashing patterns, demonstrating significant improvements in bit rate and practical bit rate compared to the traditional row column paradigm (RCP) [5, 6]. Recently, work has shown that a viable strategy of enhancing system performance is to simultaneously combine distinct optimization techniques into a singular method. For instance, Speier et al. tested the performance of a 'famous faces' stimulus paradigm integrated with a previously published particle filtering algorithm into a singular approach, establishing that the concatenation of two distinct methodologies into one offers superior results versus both approaches alone [7, 8].

This study surveys the differences in system performance between three proposed flashing patterns: Row-Column Paradigm (RCP), Checkerboard Paradigm (CBP), and the Combinatorial Paradigm (COMB) along with the integration of a language model using a particle filtering algorithm [8]. We hypothesize that the improvements offered by different flashing patterns are negligible in comparison to those from the incorporation of a language model, and therefore that improvements to BCI performance lie outside of flashing pattern optimization.

## Checkerboard paradigm

The checkerboard paradigm, CBP, was introduced as a way of improving upon the errors associated with the RCP, while concurrently improving overall BCI performance [5]. The goal with the CPB was therefore to design a novel flashing pattern that addressed the constraints associated with the RCP: the adjacency effect and the double flash pattern [9, 10]. The adjacency effect describes situations where flashes of an adjacent row or column (i.e., non-target characters) draws the user's attention, leading to false-positive P300 signals and ultimately erroneous detections of the intended character [9]. Further, the double flash pattern highlights an inadvertent conundrum associated with the RCP: random sequential row (column) or column (row) flashes can decrease the temporal resolution of the P300 signal [10]. First, because a requisite of the oddball paradigm is the presentation of "deviant stimuli" (i.e., random stimuli), consecutive flashes can impair the detection of the second flash. That is, only the first flash of the target row (column) flash will elicit the P300 signal; the second will not. Kanwisher reported this observation as the repetition blindness phenomenon [11]. In a standard rapid serial presentation (RSVP) task, consecutive stimuli presented with a temporal resolution of less than 500 milliseconds abate the recognition of the succeeding stimuli. In our P300 speller, the flash duration for a single target selection and the ISI are both 62.5 milliseconds, meaning the second flash occurs 125 milliseconds from the onset of the preceding flash, thereby diminishing the ability of the user to resolve the detection of the second flash. Hence, the aim of the CBP sought to mitigate these issues by addressing them in the stimulus design.

The CBP superimposes an imaginary checkerboard over the matrix in such a way that each adjacent character belongs to a different class [5]. Because a checkerboard inherently has an alternating pattern of two colors, the adjacent characters are grouped into two distinct classes. The characters of these two classes then randomly populate one of two corresponding virtual

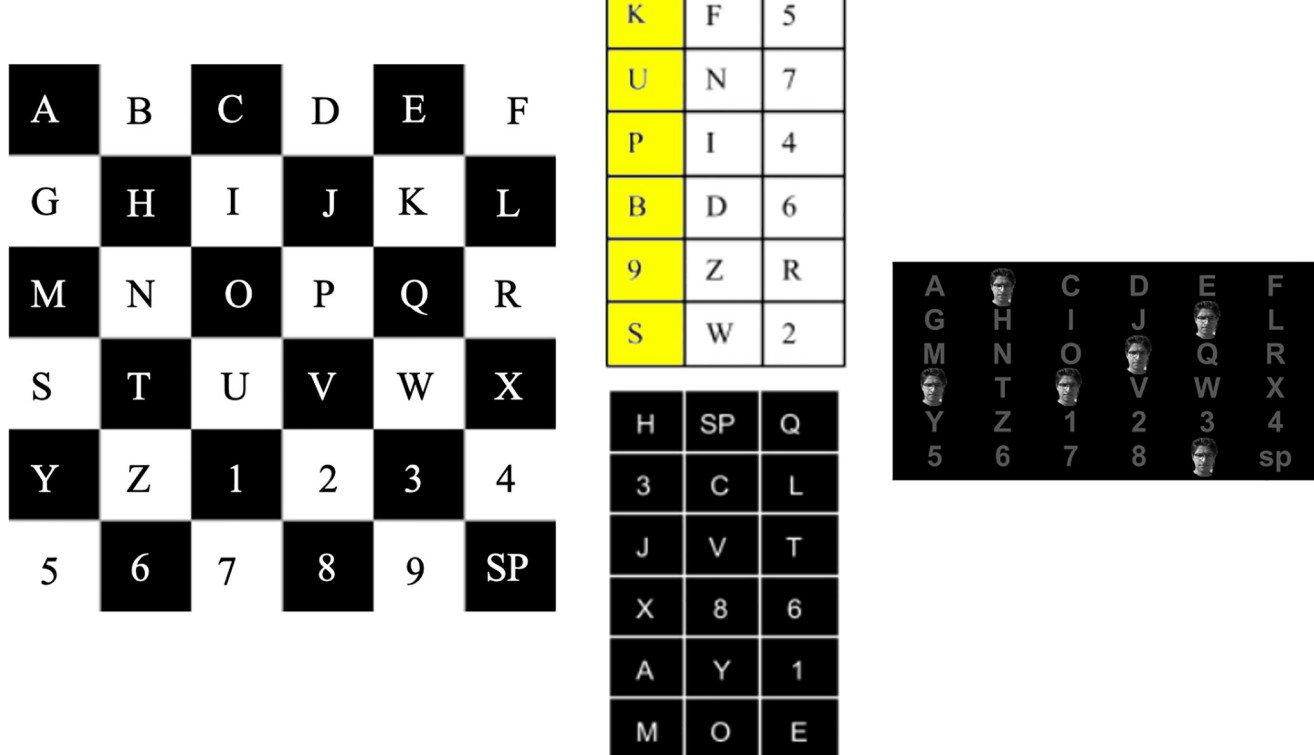

**Fig 1. Checkerboard paradigm (CBP) in a 6x6 matrix. Left:** Matrix with imaginary checkerboard superimposed over it; adjacent characters are assigned to different classes. *Center:* Characters arranged in a virtual matrix, each matrix represents a different class. In this example, the first column has been selected from the top matrix. *Right:* Actual display seen by user; a face is flashed on top of the selected characters from the virtual matrices.

matrices, which the user never observes. These virtual matrices determine the stimulus pattern for each trial (i.e., each target selection). During each target selection, the rows within both virtual matrices are flashed followed by the columns of both virtual matrices. As the rows and columns of the virtual matrices are flashed, the corresponding characters on the real matrix are presented to the user. This methodology reduces the adjacency effect, ensuring that adjacent characters never experience simultaneous flashes, and further safeguards against sequential flashes (i.e., double flashes). However, it requires a larger number of flashes in order to distinguish between each of the characters in the grid. Fig 1 depicts a schematic of the CBP.

## Combinatorial paradigm

The Combinatorial Paradigm (COMB) proposed by Jin et al. utilizes mathematical combinations to minimize the number of flashes per trial with the intention of optimizing the practical bit rate of the system [6]. Reducing the number of flashes per trial would hypothetically improve the selection rate (i.e., due to a reduced number of flashes for classification), leading to an improved PBR (practical bit rate), while still maintaining the vitality of the P300 amplitude. The goal of the COMB paradigm was therefore to optimize the number of target flashes to improve the efficacy of the system. To choose an optimal number of flashes per trial, Jin et al. used the binomial coefficient of the $x^k$ term of $(1 + x)^n$, where n equals the total number of flashes per trial, and k equals the number of flashes on the target character. A schematic of COMB is depicted in Fig 2.

**Fig 2. Combinatorial paradigm (COMB) with ($C_2^9$) flashing pattern.** *Left:* Each character is assigned a unique, two number identifier corresponding to the time it will be flashed. For example (1,3) indicates that the character will be flashed in the first and third flash. For simplicity, the characters in this figure are assigned indices sequentially, in practice the assignment would be random. In this case, we depict the third flash; so all characters corresponding to the number 3 are flashed. *Right:* The output seen by a user; a face is flashed over characters that are assigned flash index 3.

Since a traditional 6x6 matrix holds up to thirty-six characters, Jin et al. proposed using 7-flash and 9-flash patterns, which locate thirty-five and thirty-six characters, respectively. The 7-flash pattern is based on the combination ($C_3^7$), meaning that each trial has seven flashes and the target character flashes three times. Here, a single trial refers to one set of stimuli for a single target selection. Therefore, selecting the character "A" for a single trial should elicit three P300 responses in a set of seven flashes. Since the combination ($C_3^7$) equals thirty-five, the 7-flash pattern locates thirty-five characters in a traditional 6x6 matrix. On the other hand, the 9-flash pattern is based on the combination ($C_2^9$), which results in each trial having nine flashes and the target character flashing two times. This combination equals thirty-six, meaning that the corresponding flashing pattern locates thirty-six characters on the traditional 6x6 matrix. The 9-flash pattern locates the same amount of characters as the RCP flash pattern, while the 7-flash pattern locates one less. Likewise, a 12-flash pattern, which mirrors the RCP flash pattern, is modeled as "12 choose 2", where there are a total of twelve flashes for two target selections per trial—($C_2^{12}$). In comparison to the 7- and 9-flash patterns, the 12-flash pattern creates a 71.43% and 33.33% increase in the total number of flashes per trial, respectively.

## Methods

### Subjects

Ten healthy subjects (6 male, 4 female) aged 20 to 35 years old participated in this study. All subjects were cognitively viable with no noticeable neurological deficits. All subjects formally consented to participate. This study was approved by the Institutional Review Board (IRB) at UCLA.

### Data collection

EEG data were collected from a 32-cap electrode (g.GAMMAcap$^2$, Guger Technologies), and signals were amplified with two 16 channel g.tec biosignal amplifiers (Guger Technologies). Signals were sampled at 256 Hz, referenced to the left ear, grounded to AFz, and filtered using a bandpass filter from 0.1 to 60 Hz. BCI2000, a BCI-based development framework, was used for stimulus presentation and data collection [12]. Users were presented with a 6x6

**Table 1. Cross subject mean offline selection rate (SR), accuracy (ACC), and information transfer rate (ITR), for each flashing pattern and classifier.**

| Models | SR (Characters/minute) | | | ACC (%) | | | ITR (bits/minute) | | |
|---|---|---|---|---|---|---|---|---|---|
| | RCP | CBP | COMB | RCP | CBP | COMB | RCP | CBP | COMB |
| LDA | 13.54 | 13.25 | 14.15 | 98.00 | 98.50 | 99.00 | 66.99 | 66.20 | 71.54 |
| Riemann | 13.89 | 13.28 | 13.84 | 98.00 | 99.50 | 98.50 | 68.62 | 67.91 | 69.31 |
| SVM | 13.70 | 13.07 | 13.76 | 79.00 | 90.50 | 87.50 | 47.10 | 56.63 | 57.04 |
| RF | 12.00 | 11.64 | 12.41 | 92.50 | 97.00 | 99.50 | 53.36 | 56.43 | 63.38 |

matrix consisting of alphanumeric characters with 'famous faces' flashes [13]. Three distinct flashing patterns, RC, CBP, and COMB, were presented to the user to assess divergence in performance (Table 1). Each flash lasted for 62.5 ms with a 62.5 ms ISI, yielding a 125 ms stimulus onset asynchrony (SOA). Subjects completed two training sessions for each flashing pattern—creating a total of six sessions per subject. Each session consisted of 10 characters, specifically "THE QUICK" and "BROWN FOX" (including the spaces). Each of the characters corresponded to 10 sets of flashes, thus each character was flashed 20 times, with a 3.5s interval between characters.

Training data from each session was used for classification in its corresponding flashing pattern. If classification reached a significant benchmark from calibration data, online testing was performed for the trained flashing pattern. In this case, classification was appreciable for all subjects, so all subjects performed three online testing sessions. The order of online testing for each flashing pattern was randomized to dilute the effects of non-familiarity. The classification was performed using a previously established particle filtering (PF) algorithm [8].

## Language model

In this study, we use a probabilistic automata model as described by Speier et al. [8]. The model employs a directed graph that has states for each substring that starts a word in the corpus, beginning with a blank root node. Nodes are connected with directed edges to nodes that add a character to the string. For example, if the model only contained the word "CAR," it would have four states: the root node representing a blank string, "C," "CA," and "CAR." When the word "CAKE" is added to the model, it shares the root node and the "C" and "CA" states, and adds two additional states: "CAK" and "CAKE." The state "CA" then links to both the states "CAR" and "CAK." If a state represents a completed word, it will begin a new word with a link back to the root. The state "CAR," for instance, links to the root because "CAR" is a complete word, but it also is the beginning of other words so it has additional links to other states such as "CARD" or "CART." The relative frequencies of substrings in the Brown English language corpus determined transition probabilities between nodes [14]. For instance, the probability of typing the letter "R" after "CA" has already been entered is determined by dividing the number of occurrences of words that begin with "CAR" by the number of times words start with "CA" in the corpus. Similarly, the probability that a word ends and the state transitions back to the root is the ratio of the number of times that word occurs in the corpus over the number of word occurrences starting with that substring.

## Classifier

Determining the probability distribution in real time over all possible strings is computationally impractical. Instead, a PF classifier was used to estimate the distribution by sampling a batch of possible output strings. Each possible string is a particle that has a pointer to a node in

the language model. Particles independently pass through the model determined by the transition probabilities. Higher-probability ones replace low-probability particles by updating the sampling based on the EEG responses. The final distribution is estimated with the proportion of particles that point to each state after they have passed through the model.

Stepwise linear discriminant analysis (SWLDA) was used to identify signal features to include in the determination of the character that the user intended to type. At training time, ordinary least squares regression was used to predict the intended character. The model adjusts the number of features used based on their significance until a certain number of flashes occurs or it converges on a feature set. At testing time, a score for a particular character t $y_t^i$ is then given by the dot product of the feature vector and the derived features from that trial. Given a target character, the overall scores can be approximated as independent samples from a normal distribution [10].

$$
f(y_t^i | x_t) = \begin{cases} \dfrac{1}{\sqrt{2\pi\sigma_a^2}} \exp\left(\dfrac{-1}{2\sigma_a^2}(y_t^i - \mu_a)^2\right) & \text{if } x_t \in A_t^i \\[3ex] \dfrac{1}{\sqrt{2\pi\sigma_n^2}} \exp\left(\dfrac{-1}{2\sigma_n^2}(y_t^i - \mu_n)^2\right) & \text{if } x_t \notin A_t^i \end{cases} \tag{1}
$$

where $\mu_a$, $\sigma_a^2$, $\mu_n$, and $\sigma_n^2$ are the means and variances of the distributions for the attended and non-attended flashes, respectively, and $A_t^i$ is the set of characters highlighted in flash i. The conditional probability of a target at time t given the EEG signal and the previous target characters $x_{0:t-1}$ can then be found:

$$
p(x_t | y_t, x_{0:t-1}) \propto p(y_t | x_t) p(x_t | x_{0:t-1}) \propto p(x_t | x_{0:t-1}) \Pi_i f(y_t^i | x_t) \tag{2}
$$

where $p(x_n | x_{0:n-1})$ is the prior probability of character $x_n$ given the previously selected characters, derived from the language model. The distribution over all output strings is approximated by particle filtering. In PF, we first generate a fixed number of samples, or "particles" to estimate the distribution. Each individual particle $j$ consists of 4 main elements: a pointer to a state in the language model $x_t^{(j)}$; a sequence that represents the different states in the history of the particle $x_{0:t}^{(j)}$; an index $m$ that denotes the most recent time point when the particle was at the root node; and an associated weight for the particle, denoted $w^{(j)}$. At initialization, we generate $P$ particles pointing to the root node, no history, and a uniform weight $\frac{1}{P}$. When a new character begins, a character $x_t^{(j)}$ is sampled for each particle from its proposal distribution. This is defined by the transition probabilities of the language model and the particle's history $x_{0:t-1}^{(j)}$.

$$
x_t^{(j)} \sim p(x_t | x_{0:t-1}^{(j)}) \tag{3}
$$

where $p(x_t | x_{0:t-1}^{(j)})$ is provided by the language model as in Eq 1.

After each stimulus-response, the score for that response, $y_t^i$, is computed and the probability weight is updated for each of the particles:

$$
w_t^{(j)} \propto p(y_t | x_t^{(j)}) \propto \Pi_i f(y_t^i | x_t) \tag{4}
$$

where $f(y_t^i | x_n^{(j)})$ is computed as in Eq 2. The weights are then normalized and the probability of an output string is found by summing the weights of all particles that correspond to that

string.

$$p(x_{0:t}|y_{1:t}) = \sum_k w_t^{(k)} \delta_{x_{0:t}}^{x_{0:t}^{(k)}}$$

(5)

where $\delta$ is the Kronecker delta. Dynamic classification was implemented by setting a threshold probability, $p_{\mathrm{thresh}}$, to determine when a decision should be made. The program flashes characters until either the maximum probability exceeds the threshold, or the number of sets of flashes reached the maximum (10 flashes). The classifier then selects the string that satisfied $\underset{x_{0:t}}{\mathrm{argmax}}\, p(x_{0:t}|y_{1:t})$. If there is a difference between this output and the previous output, the older characters are treated as errors and are replaced. A new batch of particles is sampled from the current particles based on their weights. Each new particle is given the same uniform weight as before. The subject moves to the next character and the process repeats. Online optimization of $p$ is impractical, so all trials use a previously reported value of 0.95 [8].

## Predictive spelling

In the adjusted model with PS, the same language model and classifier are used, but the projection step is altered to estimate the probabilities for complete words. A subset of particles, *rho*, continues to the root node during projection. Since particles can move multiple steps in a single transition phase, the particle history can be greater than $t$, and we denote it as $n_j$. After projection, the probability distribution for complete words is calculated by summing the weights of the relevant projected particles.

The top $k$ words are inserted into specific positions in the character grid. The EEG signals associated with the flashing of those specific cells are applied to particles that are mapped to those words. Particles mapped to less likely words are assigned a probability of 0 and are replaced in the next step of the algorithm. The probability of a complete word selection was determined empirically and set to 0.40. At a given point, the user was presented with six word suggestions.

## Evaluation

The performance of a BCI system is based on the balance between its ability to perform a particular task and the time it takes to achieve the goal. In lieu of this tradeoff, evaluation is commonly based on the bit rate (BR).

$$\mathrm{BR} = \sum_z p(z) \sum_x p(x|z) log \frac{p(x|z))}{p(x)}$$

(6)

The most common use of the BR is information transfer rate (ITR). We assume a uniform distribution across all the characters $p(x) = \frac{1}{N}$ (where N is 36, the size of the alphabet). The same assumption applies to errors so

$$p(x|z) = \begin{cases} ACC_c & x = z \\ \dfrac{1 - ACC_c}{N - 1} & x \neq z \end{cases}$$

(7)

where $ACC_c = \frac{\sum_t \delta_{x_t}^{z_t}}{n}$ is the single character accuracy and n is the total number of characters

selected. This reduces the bit rate to

$$BR = logN + ACC_c \ logACC_c + (1 - ACC_c) \ log\frac{1 - ACC_c}{N - 1} \qquad (8)$$

This is then multiplied by the average number of characters selected per minute (selection rate) to produce the ITR.

$$ITR = BR * CPM \quad (11)$$

Evaluating predictions at a character level is not reasonable in this predictive spelling (PS) scheme, as sentences with incorrect words could be a different length from the target sentence. In order to circumvent this, accuracy is based on Levenshtein distance (LD) [15]. We then have $ACC_c = \frac{n - LD(x,z)}{n}$, and the equations above hold.

Because the distributions for the metrics used are not normally distributed, significance was tested using the nonparametric Kruskal-Wallis test.

## Results

### Offline analysis

In the offline analysis, no one paradigm significantly outperformed any other across the three measured metrics. Table 2 shows the offline selection rate (in characters per min), accuracy, and ITR for each flashing pattern. The differences in median SR between the three flashing paradigms were not found to be statistically significant (H = 2.96, p = 0.227). The accuracy across the different paradigms, while high, are also not significantly different (H = 0.581, 0.748). While the combinatorial paradigm has a slightly better accuracy, there are no significant differences in ITR between the three paradigms (H = 5.257, p = 0.0722).

We repeated the offline analysis using two subsets of the training data, one with half the data (5 characters) and the other with 30% (3 characters). When reducing the training data by half, the ITR values for the RC, CBP, and COMB paradigms decreased by 7.34%, 2.06%, and 3.57%, respectively. None of these decreases was statistically significant (p = 0.13, 0.41, and p = 0.19, respectively). When decreasing the training data to 30%, the three ITR values decreased by 16.13%, 3.29%, and 9.81%. The CBP performance was not significantly different from the full dataset (p = 0.28), but the RC and COMB paradigms did significantly decrease (p = 0.0025 and p = 0.0024, respectively). When using 50% of the dataset, the COMB

**Table 2. The offline selection rate (SR), accuracy (ACC), and information transfer rate (ITR), for each flashing pattern.**

| | SR (Characters/minute) | | | ACC (%) | | | ITR (bits/minute) | | |
|---|---|---|---|---|---|---|---|---|---|
| Subjects | RCP | CBP | COMB | RCP | CBP | COMB | RCP | CBP | COMB |
| 1 | 13.46 | 13.87 | 13.91 | 100.00 | 95 | 100.00 | 69.61 | 64.19 | 71.93 |
| 2 | 13.32 | 12.94 | 15.07 | 95 | 100.00 | 100.00 | 61.61 | 66.89 | 77.91 |
| 3 | 14.769 | 13.75 | 15.71 | 100.00 | 100.00 | 95 | 76.35 | 71.10 | 72.70 |
| 4 | 11.12 | 9.80 | 13.79 | 100.00 | 100.00 | 100.00 | 57.51 | 50.64 | 71.31 |
| 5 | 12.938 | 13.52 | 12.97 | 95 | 95 | 100.00 | 59.86 | 62.56 | 67.07 |
| 6 | 14.90 | 14.44 | 14.55 | 95 | 100.00 | 100.00 | 68.97 | 74.63 | 75.20 |
| 7 | 14.747 | 14.24 | 13.87 | 100.00 | 100.00 | 100.00 | 76.24 | 73.63 | 71.72 |
| 8 | 13.278 | 12.70 | 13.93 | 100.00 | 100.00 | 100.00 | 68.64 | 65.65 | 72.03 |
| 9 | 14.307 | 12.57 | 14.14 | 95 | 100.00 | 95 | 66.20 | 64.96 | 65.42 |
| 10 | 12.55 | 14.63 | 13.56 | 100.00 | 95 | 100.00 | 64.87 | 67.71 | 70.10 |
| Mean | 13.54 | 13.25 | 14.15 | 98 | 98.5 | 99 | 66.99 | 66.20 | 71.54 |

paradigm's performance was still significantly better than RC and CB (p = 0.02 and p = 0.03, respectively). When using 30% of the dataset, the difference between COMB and RC remained significant (p = 0.008), but the difference with CBP was no longer significant (p = 0.42). When using the reduced training, the optimal method varied across subjects, with one subject performing best with RC, four performing best with CBP, and 5 performing best with COMB.

Different classifiers were also analyzed to assess if there were any superior approaches to LDA [16]. We explored a support vector machine (SVM), random forest (RF), and a Riemannian geometry classifier implemented using the covariancetoolbox (https://github.com/alexandrebarachant/covariancetoolbox) package, modified as in Barachant et al [17]. We found that LDA achieves a significantly better ITR across all three flashing paradigms compared to SVM (p = 0.0015, 0.0208, and 0.0354, respectively) and RF (p = 0.0022, 0.0018, 0.0018, respectively). The Riemannian geometry classifier achieved statistically similar ITR to LDA (p = 0.34, p = 0.41, and p = 0.15, respectively). The COMB flashing paradigm had the highest ITR when using each of these classifiers. However, the difference was only statistically significant using the RF classifier (p = 0.002 and p = 0.007 compared to RC and CBP, respectively). No significant difference was found between flashing paradigms when using the SVM (p = 0.47 and 0.08, respectively) or Riemannian geometry classifiers (p = 0.20 and 0.32, respectively).

## Online analysis

Table 3 shows the online selection rate (in characters per min), accuracy, and ITR for each flashing pattern. Despite the RCP yielding the highest mean SR, there were no significant differences in the mean SR for any of the flashing patterns (H = 0.98, p = 0.611) (Table 1). In contrast, the difference in the median accuracy across the flashing patterns was found to be statistically significant (H = 7.399, p = .025). Pairwise Mann-Whitney tests between each of the flashing patterns demonstrate that the CBP pattern was significantly higher than the COMB flashing pattern (p = .0045). There are no appreciable differences in accuracy between RCP and CBP across (p = .271) and between RCP and COMB for (p = .112). Further, the mean ITR, which is a function of both accuracy and SR, was not significantly different for any of the flashing patterns (H = 1.46, p = 0.481), consistent with SR and the offline paradigm. Therefore, no appreciable differences were detected in BCI performance among each of the flashing patterns.

**Table 3. The online selection rate (SR), accuracy (ACC), and information transfer rate (ITR), for each flashing pattern.**

| Subjects | SR (Characters/minute) | | | ACC (%) | | | ITR (bits/minute) | | |
|---|---|---|---|---|---|---|---|---|---|
| | RCP | CBP | COMB | RCP | CBP | COMB | RCP | CBP | COMB |
| 1 | 14.65 | 15.08 | 15.86 | 100.00 | 100.00 | 97.27 | 75.73 | 77.94 | 76.91 |
| 2 | 16.46 | 16.11 | 17.14 | 100.00 | 100.00 | 100.00 | 85.11 | 83.3 | 88.6 |
| 3 | 16.79 | 15.43 | 16.61 | 100.00 | 100.00 | 100.00 | 86.82 | 79.76 | 85.89 |
| 4 | 12.38 | 11.48 | 10.63 | 98.59 | 100.00 | 94.74 | 61.77 | 59.36 | 48.93 |
| 5 | 15.05 | 13.89 | 12.41 | 81.25 | 100.00 | 82.50 | 52.87 | 71.79 | 44.71 |
| 6 | 15.34 | 13.63 | 14.72 | 100.00 | 100.00 | 100.00 | 79.3 | 70.48 | 76.11 |
| 7 | 15.35 | 13.19 | 12.77 | 100.00 | 100.00 | 98.57 | 79.35 | 68.21 | 63.7 |
| 8 | 13.39 | 12.58 | 14.72 | 95.95 | 100.00 | 76.92 | 63.17 | 65.06 | 47.2 |
| 9 | 13.71 | 14.51 | 14.79 | 100.00 | 100.00 | 90.67 | 70.88 | 75.02 | 62.78 |
| 10 | 12.79 | 12.18 | 10.01 | 100.00 | 100.00 | 97.67 | 66.13 | 62.99 | 48.97 |
| Mean | 14.59 | 13.81 | 13.97 | 97.58 | 100.00 | 93.83 | 72.11 | 71.39 | 64.38 |

## Waveform analysis

The P300 signal for each flashing pattern was evaluated at CPz, POz, PO7, and PO8 to examine for meaningful differences in the amplitudes of the waveforms [18]. Stimulus responses during online sessions were grouped based on whether the stimulus contained the target character. The average attended and non-attended responses were calculated for each subject and a global average was produced across subjects for each channel. Significance was tested at each latency to determine whether attended and non-attended responses differed significantly using Wilcoxon signed rank tests correcting for multiple comparisons using false discovery rate.

For all three stimulus paradigms, there were significant differences between attended and non-attended stimulus responses. Each of the four channels had a large positive peak preceded by a smaller negative peak in the attended responses. In the parieto-occipital channels, the negative peak occurred at a latency of approximately 200 ms and the positive peak at a latency of 300 ms. In the CPz channel, these peaks were slightly later, occurring at approximately 300 ms and 400 ms latencies, respectively. In the CPz, POz, and PO8 channels, the positive peak was significantly different from the non-attended response. The peak in the PO7 channel was not statistically significant, most likely because of a high variance across subjects.

The average signals were compared across the responses for the three stimulus paradigms. While the positive peaks for the parieto-occipital channels were generally larger for the checkerboard paradigm, no significant trend was seen between the three groups. This result suggests that the stimulus paradigm does not significantly affect the stimulus response produced.

The peak amplitude for each subject ranged from 200 ms to 500 ms, and thus amplitudes of the P300 signal were only compared within subjects. For subject 1, there were no significant differences in the amplitude of the P300 signal at any electrode location ($F_{2,195} = 2.75$, p = 0.066; $F_{2,195} = 0.81$, p = 0.445; $F_{2,192} = 2.72$, p = 0.068; $F_{2,192} = 0.60$, p = 0.548). The P300 signal peaked around 400 ms for each flashing pattern in this subject with similar latencies for each flashing pattern. However, in subject 2, there were significant differences in the amplitude at CPz and PO7 ($F_{2,99} = 7.45$, $p < 0.05$; F2,81 = 9.72, $p < 0.05$), albeit at POz and PO8 there were no significant differences in the amplitude of the waveforms ($F_{2,84} = 0.89$, p = 0.416; $F_{2,195} = 1.89$, p = 0.158) (Fig 3). Since the mean selection accuracies for each flashing pattern for subject 2 was 100%, the P300 responses represent pure responses undiluted by incorrect selections. Interestingly, subject 3 also had a mean selection accuracy of 100% for each flashing pattern, yet only PO8 demonstrated a significant difference in the mean amplitude of the P300 signal ($F_{2,90} = 3.1849$, $p < 0.05$), suggesting that the amplitude of the P300 signal is not a function of the flashing pattern, but of some psychological variable, such as attention or motivation.

## Discussion

A robust, clinically viable BCI speller requires high accuracy ($>$90%), and speed (at least 15-19 characters per minute) [19]. Although the functional utility of the P300 speller has been demonstrated in invasive conditions, specifically with signals acquired with electrocorticography (ECoG), the long-term safety and utility has yet to be determined. In order to ameliorate the risks of an invasive procedure, several studies aim to optimize the utility of a P300 speller with a non-invasive, EEG-based paradigm. Much work has been done to try and optimize the flashing pattern used, but has yielded mixed results [5, 6].

Our study aimed to provide a meaningful, standardized comparison of performance for each flashing pattern, incorporating optimization methodologies that have been shown to enhance performance. In our study, alternative flashing paradigms did not significantly

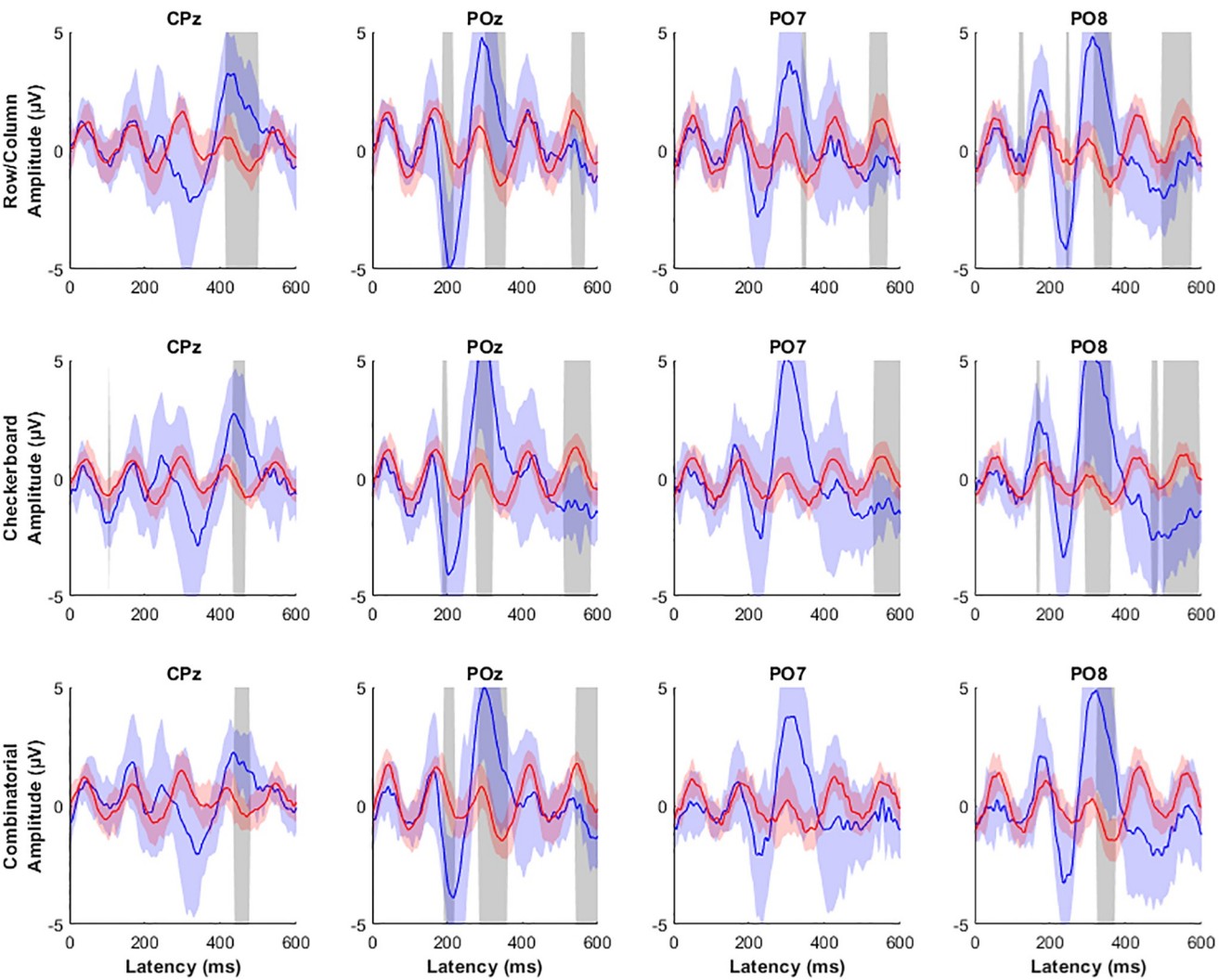

**Fig 3. Target P300 waveforms.** The average target response for each flashing pattern at CPz, POz, PO7, and PO8 for subject 2 when using the row/column (blue), checkerboard (green), and combinatorial (red) flashing paradigms.

improve typing performance in a system with dynamic stopping and language model priors. The mean online selection rate, mean online accuracy, and mean online ITR were not significantly different for any of the three flashing patterns. This observation contrasts with reports from both Townsend (2010) and Jin (2010) that the traditional RCP flashing pattern failed to meet equivalent performance standards compared to the CBP and COMB flashing patterns, respectively [5, 6].

Townsend (2010) reported that the CBP flashing pattern yielded both a greater online accuracy and practical bit rate [5]. In a 72 character grid, there are 24 flashes per target selection in the CBP flash pattern compared 17 flashes per target selection the RCP flash pattern. Because the number of flashes in the CBP is higher than in the RCP, this would naturally lead to a greater time duration for each target selection, leading to lower SR. Leveraging dynamic stopping, where the number of flashes per target selection is modulated by the classification threshold, would dilute this disparity, normalizing the selection rate for the CBP and RCP flashing patterns. While we find that the CBP pattern has significantly higher accuracy, we hypothesize

that the excellent accuracy performance of the CBP pattern in the online paradigm is due to the fact that CBP optimizes for accuracy while making significant concessions to speed.

Jin (2010) stated that mean offline practical bit rate was significantly different for the 9-flash pattern compared to the 12-flash pattern (RCP), as a result of the diminished number of flashes required for a character selection—a 33.33% decrease in the number of flashes per selection [6]. Although there were fewer characters required for each character selection, this did not necessarily translate to a higher online selection rate.

Our results suggest that dynamic stopping, where the number of flashes per target selection changes as a function of a classification threshold needed to select a character, reduces the performance effects of a nine-flash pattern with static number of flashes. Dynamic stopping allows the system make decisions without needing to wait for a required number of flashes, thereby reducing the impact of the flashing pattern on performance.

One factor that was not addressed in this study would be the effect of using a multi-stage approach that allows for either a reduction in the number of targets on screen, or an increase in the number of potential targets [20]. In this case, the targets on screen still need to be highlighted, which could be done using any of the methods presented in this article (or others such as single character flashing). The requirement of making multiple decisions for each character selection could prioritize accuracy, possibly making the CBP preferable in this context. Future work should investigate how these flashing paradigms would interact with a multi-stage approach.

## Conclusion

This study shows that when used in conjunction with other established methods, proposed flashing paradigms do not make a significant impact on P300 speller performance. A large contributing factor to this phenomenon could be that dynamic stopping allows the system to make decisions without needing to wait for a required number of flashes, reducing the impact of the flashing paradigm. This result likely implies that current bottlenecks in P300 speller performance lie outside the type of flashing paradigm used, and that optimization methods should be focused on improvements to language models and predictive spelling.

## Author Contributions

**Conceptualization:** Corey Arnold, Nader Pouratian, William Speier.

**Data curation:** Shrita Pendekanti, Robert Tran, William Speier.

**Formal analysis:** Robert Tran, Saarang Panchavati, Corey Arnold, William Speier.

**Investigation:** Nand Chandravadia, Shrita Pendekanti, Dustin Roberts, William Speier.

**Methodology:** Nand Chandravadia, Dustin Roberts, William Speier.

**Project administration:** William Speier.

**Resources:** Nader Pouratian.

**Software:** William Speier.

**Supervision:** Corey Arnold, Nader Pouratian, William Speier.

**Validation:** Nand Chandravadia, Dustin Roberts, Robert Tran, Saarang Panchavati, Nader Pouratian, William Speier.

**Writing – original draft:** Nand Chandravadia, Dustin Roberts, Nader Pouratian, William Speier.

**Writing – review & editing:** Nand Chandravadia, Shrita Pendekanti, Dustin Roberts, Robert Tran, Saarang Panchavati, Corey Arnold, Nader Pouratian, William Speier.

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
