## [Decision Letter · Decision Letter 0]

20 Feb 2023

PONE-D-22-19793Comparing P300 flashing paradigms in online typing with language modelsPLOS ONE

Dear Dr. Speier,

Thank you for submitting your manuscript to PLOS ONE. After careful consideration, we feel that it has merit but does not fully meet PLOS ONE’s publication criteria as it currently stands. Therefore, we invite you to submit a revised version of the manuscript that addresses the points raised during the review process.

The presented study design and results must be sufficient to support the claims made.

As recommended by reviewers, the authors should run the analysis multiple times, using

multiple different classification algorithms that have achieved the highest performances on this type of data. Either these control variables should be added to the analyses in the manuscript, or the general claim about the impact of flashing patterns should be constrained to only apply to the given algorithm and amount of training data. For completeness, the paradigm comparison should contain another class of flashing patterns/P300 interfaces. Multi-stage approaches, where first one of a few targets representing a group of characters is selected, and afterward a second selection is made for the exact characters, should also be studied. Due to the lower number of targets, P300 amplitude, and classifier performance might be increased, and the crowding effect is decreased. This might turn out to significantly impact the evaluated metrics, with or without language model.

Include the ethical approval reference number.

There are serious issues with the description of the methods: evaluation of some parameters (BR, MI) is described, but parameters are not used, and vice versa some parameters (SR) are used but not described.

The description of the Data collection is insufficient. According to the authors, each subject completes three training sessions for each flashing paradigm, in which each subject copied three, ten-character words. No information is provided regarding the offline training session and the online one. How many training sessions correspond to the offline study and the online study? How many characters are used in each session? 30 characters? If the results are based on the offline and online sessions, the description of the methodology used in both types of session should be very clear. In another side, it is not justified the used of “famous faces” flashes.

If one of the contribution of the paper is the use of dynamic stopping to reduce the number of flashes required to select a character, it had been important to include, in the result section, information regarding the number of flashes required in the offline and online section for each paradigm. This information could explain the difference obtained between different ITR and SR for the same value of ACC.

Other issues:

- In page 2 line 60-62 the authors state: « In the P300 speller, the flash duration for a single target selection and the ISI are both 62.5 milliseconds »---Do you mean that any P300 speller uses this timing ? This is not correct.

- In the language model section the authors mention Figure 2 (line 135) however Figure 2 corresponds to the COMB pattern.

- In the Predictive Spelling section the authors mention Figure 1 (line 224) however Figure 1 corresponds to CBP pattern.

- Figure 3 is not mention into the text

We look forward to receiving your revised manuscript.

Kind regards,

Gennady S. Cymbalyuk, Ph.D.

Academic Editor

PLOS ONE

Journal Requirements:

2. Please ensure that you have specified (1) whether consent was informed and (2) what type you obtained (for instance, written or verbal, and if verbal, how it was documented and witnessed). If your study included minors, state whether you obtained consent from parents or guardians. If the need for consent was waived by the ethics committee, please include this information.

“This material is supported in part by the National Science Foundation Graduate Research Fellowship Program (DGE-2034835).”

Reviewers' comments:

Reviewer's Responses to Questions

**Comments to the Author**

1. Is the manuscript technically sound, and do the data support the conclusions?

Reviewer #1: Partly

Reviewer #2: Partly

2. Has the statistical analysis been performed appropriately and rigorously? 

Reviewer #1: Yes

Reviewer #2: Yes

3. Have the authors made all data underlying the findings in their manuscript fully available?

Reviewer #1: Yes

Reviewer #2: No

4. Is the manuscript presented in an intelligible fashion and written in standard English?

Reviewer #1: Yes

Reviewer #2: Yes

5. Review Comments to the Author

Reviewer #1: The data onlt partly supports the conclusions:

By stating that flashing patterns do not form the bottleneck of P300 spellers,

this manuscript produces an insightful conclusion that is very useful for

deciding where the focus of future studies should lie. In general, the evidence

presented supports this conclusion. However, some controlled variables are

lacking to be able to claim that this conclusion generally holds. First, the

performance of different flashing patterns should be studied in the presence of

limited training data, by running the offline analysis multiple times with

different amounts of training data. The limited training data case is often of

interest for BCI design, since the calibration time of a practical BCI should

be as short as possible. It might very well be the case that for certain

training set sizes, the flashing patterns do have a significant impact on the

evaluated metrics, with or withouth the language model. Second, it might also

be the case that a significant effect is observed when using another

classification algorithm than SWLDA. The authors only evaluate the performance

metrics using a SWLDA classifier, but it is known that more performant P300 classifiers do

exist [1]. A more performant classifier could increase the impact of the decoded

brain response relative to the impact of the language model, hence causing

significant differences across flashing patterns. In order to make the claim

more convincing, the authors should run the analysis multiple times, using

multiple different classification algorithms that have achieved the highest

performances on this type of data. Either these these control variables should

be added to the analyses in the manuscript, or the general claim about the

impact of flashing patterns should be constrained to only apply to the given

algorithm and amount of training data.

1. Lotte, F., Bougrain, L., Cichocki, A., Clerc, M., Congedo, M., Rakotomamonjy, A., & Yger, F. (2018). A review of classification algorithms for EEG-based brain–computer interfaces: a 10 year update. Journal of neural engineering, 15(3), 031005.

Other comments:

For completeness, the paradigm comparison should contain another class of

flashing patterns/P300 interfaces. Multi-stage approaches, where first one of

a few targets representing a group of characteres is selected, and afterwards

a second selection is made for the exact characters, should also be studied.

Due to the lower number of targets, P300 amplitude and classifier performance

might be increased, and the crowding effect is decreased. This might turn out

to significantly impact the evaluated metrics, with or without language

model. A multi-stage approach like this has been implemented by [2].

Include the ethical approval reference number.

2. Treder, M. S., & Blankertz, B. (2010). (C) overt attention and visual speller design in an ERP-based brain-computer interface. Behavioral and brain functions, 6(1), 1-13.

Reviewer #2: In this paper, the authors compare the typing speeds of three different flashing paradigms in a P300 speller : the Row-Column (RCP), Checkerboard (CBP), and Combinatorial Paradigms (COMB). The comparative results are based on performance in online and offline setting, specifically, performance based on accuracy (ACC), selection rate (SR) and Information transfer rate (ITR).

The RCP paradigm is the most used paradigm, and the others two paradigms have also yet proposed in other studies ([5] and [6] for CBP and COMB respectively). In this sense, a comparison between these paradigms are not a really innovative contribution, since this comparison has been carried out in the different studies. Besides, the obtained results in this study do not show significant differences and then do not provide significant progress.

In my opinion, the paper is difficult to follow. The description of the methodology is very confused. The description of the evaluation section is very difficult to understand. The authors describe many equations, but, finally, the parameters obtain from these equations are not used in the study: Bit rate (BR), BR’, Mutual information (MI). The only parameters used in the results sections are the Accuracy (Acc), the ITR and the Selection Rate (SR) so all the description provided in the evaluation section is very confused. Regarding the Selection Rate (SR), it is not explained how to obtain it.

The description of the Data collection is really insufficient. According to the authors each subject completes three training sessions for each flashing paradigm, in which each subject copied three, ten character words. No information is provided regarding the offline training session and the online. How many training sessions correspond to the offline study and the online study? How many characters are used in each session? 30 characters? If the results are based on the offline and online sessions, the description of the methodology used in both types of session should be very clear. In another side, it is not justified the used of “famous faces” flashes.

If one of the contribution of the paper is the use of dynamic stopping to reduce the number of flashes required to select a character, it had been important to include, in the result section, information regarding the number of flashes required in the offline and online section for each paradigm. This information could explain the difference obtained between different ITR and SR for the same value of ACC.

Other issues:

- In page 2 line 60-62 the authors state: « In the P300 speller, the flash duration for a single target selection and the ISI are both 62.5 milliseconds »---Do you mean that any P300 spller uses these timing ? This is not correct.

- In the language model section the authors mention the Figure 2 (line 135) however the Figure 2 corresponds to the COMB pattern.

- In the Predictive Spelling section the authors mention the Figure 1 (line 224) however the Figure 1 corresponds to CBP pattern.

- Figure 3 is not mention into the text

6. PLOS authors have the option to publish the peer review history of their article (what does this mean?). If published, this will include your full peer review and any attached files.

Reviewer #1: No

Reviewer #2: No

---

## [Author Response · Author response to Decision Letter 0]

3 Oct 2023

Reviewer #1: 

By stating that flashing patterns do not form the bottleneck of P300 spellers, this manuscript produces an insightful conclusion that is very useful for deciding where the focus of future studies should lie. In general, the evidence presented supports this conclusion. 

We appreciate the reviewer’s supportive comments.

However, some controlled variables are lacking to be able to claim that this conclusion generally holds. First, the performance of different flashing patterns should be studied in the presence of limited training data, by running the offline analysis multiple times with different amounts of training data. The limited training data case is often of interest for BCI design, since the calibration time of a practical BCI should be as short as possible. It might very well be the case that for certain training set sizes, the flashing patterns do have a significant impact on the evaluated metrics, with or without the language model. 

We thank the reviewer for this suggestion. We repeated the offline analysis using two subsets of the training data, one with half the data (5 characters) and the other with 30% (3 characters). When reducing the training data by half, the ITR values for the RC, CBP, and COMB paradigms decreased by 7.34%, 2.06%, and 3.57%, respectively. None of these decreases was statistically significant (p=0.13, 0.41, and p=0.19, respectively). When decreasing the training data to 30%, the three ITR values decreased by 16.13%, 3.29%, and 9.81%. The CBP performance was not significantly different from the full dataset (p=0.28), but the RC and COMB paradigms did significantly decrease (p=0.0025 and p=0.0024, respectively). Even with the significant decrease, the COMB paradigm still had the highest average ITR (64.52 vs. 56.18 for RC and 64.02 for CBP). When using the reduced training, the optimal method varied across subjects, with one subject performing best with RC, four performing best with CBP, and 5 performing best with COMB. The results of this analysis have been added to the offline analysis section.

Second, it might also be the case that a significant effect is observed when using another classification algorithm than SWLDA. The authors only evaluate the performance metrics using a SWLDA classifier, but it is known that more performant P300 classifiers do exist [1]. A more performant classifier could increase the impact of the decoded brain response relative to the impact of the language model, hence causing significant differences across flashing patterns. In order to make the claim more convincing, the authors should run the analysis multiple times, using multiple different classification algorithms that have achieved the highest performances on this type of data. Either these control variables should be added to the analyses in the manuscript, or the general claim about the impact of flashing patterns should be constrained to only apply to the given algorithm and amount of training data.

1. Lotte, F., Bougrain, L., Cichocki, A., Clerc, M., Congedo, M., Rakotomamonjy, A., & Yger, F. (2018). A review of classification algorithms for EEG-based brain–computer interfaces: a 10 year update. Journal of neural engineering, 15(3), 031005.

We thank the reviewer for this suggestion. While the SWLDA algorithm is the most common algorithm used for P300 analysis there have been multiple studies testing other algorithms with mixed results. We have added an analysis using several of the algorithms mentioned in the suggested review in our offline analysis. Overall, no algorithm produced superior results to SWLDA, with the Riemannian geometry classifier producing equivalent results and random forest and SVM classifiers performing significantly worse. The results of this analysis are included in the offline results section of the results.

For completeness, the paradigm comparison should contain another class of flashing patterns/P300 interfaces. Multi-stage approaches, where first one of a few targets representing a group of characters is selected, and afterwards a second selection is made for the exact characters, should also be studied. Due to the lower number of targets, P300 amplitude and classifier performance might be increased, and the crowding effect is decreased. This might turn out to significantly impact the evaluated metrics, with or without language model. A multi-stage approach like this has been implemented by [2].

2. Treder, M. S., & Blankertz, B. (2010). (C) overt attention and visual speller design in an ERP-based brain-computer interface. Behavioral and brain functions, 6(1), 1-13.

We appreciate the reviewer’s suggestion. However, the addition of another arm in the study is not feasible since data collection is already completed. Each subject in this study completed all three arms in the same session to reduce session effects and to allow for randomization of the order of the arms. Adding in another arm would introduce uncontrolled variables, even if we were able to recruit the same subject population for the study (which is unlikely since several of the subjects were students who have since graduated and left the university).

We also argue that a multi-stage approach is a different (although related) concept from flashing paradigm. As the reviewer states, a multi-stage approach allows for either a reduction in the number of targets on screen, or an increase in the number of potential targets. The targets on screen still need to be highlighted in some manner, which could be done using any of the methods presented in this article (or others such as single character flashing). It would be interesting to see how a multi-stage approach would interact with the different flashing paradigms (as well as other improvements such as language models), however, that is outside the scope of this study. We instead include a discussion about how a multi-stage approach could impact the results we saw as well as a reference to the suggested paper in the discussion section.

Include the ethical approval reference number.

The IRB number has been included.

Reviewer #2: 

The RCP paradigm is the most used paradigm, and the others two paradigms have also yet proposed in other studies ([5] and [6] for CBP and COMB respectively). In this sense, a comparison between these paradigms are not a really innovative contribution, since this comparison has been carried out in the different studies. 

The reviewer is correct that each of these stimulus paradigms has been previously studied, so the novelty of this study does not lie in the usage of these paradigms. In the studies that have investigated these paradigms, other advances have not been incorporated, so it has been unclear how they would interact and whether their performance improvements would hold in the presence of other model improvements such as famous face flashing, dynamic stopping, or language model integration. These methods are designed to have similar effects to the CBP and COMB flashing paradigms (i.e., improved accuracy for CBP and improved selection speed for COMB). It was not clear whether the methods would synergize and lead to improved results, or whether the flashing paradigms would be redundant with the other methods.

In addition, there are several minor components to the study that haven’t been explored previously:

1. While the CBP and COMB flashing paradigms were both previously compared to RCP, they have not been compared to each other.

2. The CBP and COMB paradigms were previously implemented and tested in much larger character grids and were not previously implemented in the standard 6x6 grid.

3. While the CBP and COMB paradigms have been previously established, much of the validation of these methods has been in offline studies, with relatively limited evaluation in online free spelling scenarios.

Besides, the obtained results in this study do not show significant differences and then do not provide significant progress.

We respectfully disagree that the lack of significant differences indicates a lack of progress. Negative results can have a significant impact as they can rebut common assumptions and suggest future directions for research. In this case, we show that changing flashing paradigms has minimal impact on typing performance when combined with other system improvements, suggesting that the focus of future studies may be better placed elsewhere (as noted by reviewer 1). However, we also show that the accuracy using the CB paradigm was perfect across all subjects in our online setting, so it could be preferable even if it doesn’t necessarily yield significantly higher ITR.

In my opinion, the paper is difficult to follow. The description of the methodology is very confused. 

The text in the methods section has been extensively edited and simplified to remove unnecessary description and details that have been published in previous papers.

The description of the evaluation section is very difficult to understand. The authors describe many equations, but, finally, the parameters obtained from these equations are not used in the study: Bit rate (BR), BR’, Mutual information (MI). The only parameters used in the results sections are the Accuracy (Acc), the ITR and the Selection Rate (SR) so all the description provided in the evaluation section is very confused. Regarding the Selection Rate (SR), it is not explained how to obtain it.

We Thank the reviewer for pointing out this confusion. Several of the descriptions of metrics were held over from a previous version of the manuscript. We apologize for this mistake. We have adjusted this section to limit it to the metrics that are used in this study.

The description of the Data collection is really insufficient. According to the authors each subject completes three training sessions for each flashing paradigm, in which each subject copied three, ten character words. No information is provided regarding the offline training session and the online. How many training sessions correspond to the offline study and the online study? How many characters are used in each session? 30 characters? If the results are based on the offline and online sessions, the description of the methodology used in both types of session should be very clear.

We thank the reviewer for pointing out some confusing elements in this section. In particular, we erroneously stated that there were three trials for each flashing paradigm when there were only two, which led to some of the confusion. The data collection section has been modified to make things clearer. We’ll include a description of the data collection below as well:

Each subject underwent two trials for each flashing paradigm during the training session. Each trial consisted of ten characters, so the training session consisted of a total of 20 characters for each of the flashing paradigms. Offline evaluation was performed across these 30 characters. During online trials, the trained models were used by subjects to type as much of a sentence of their choosing in five minutes. The number of characters therefore varied across subjects with a range between 50 and 85. 

In another side, it is not justified the used of “famous faces” flashes.

Famous faces flashing has been shown previously to have a significant impact on P300 decoding (Kaufmann, 2011). We have previously tested it in our system and in the presence of a language model and demonstrated similar improvements (Speier, 2017). 

If one of the contribution of the paper is the use of dynamic stopping to reduce the number of flashes required to select a character, it had been important to include, in the result section, information regarding the number of flashes required in the offline and online section for each paradigm. This information could explain the difference obtained between different ITR and SR for the same value of ACC.

Dynamic stopping is directly related to the selection rate (SR). If dynamic stopping were not employed, all subjects would have the same selection rate. The amount of time required to type one character would be the number of seconds required to present 10 sets of 12 flashes (10*12*125ms) plus the time delay between characters (3.5s) for a total of 18.5 seconds. The SR would then be 60/18.5=3.24 characters/minute and the ITR would similarly be reduced since it is the product of SR and bit rate.

The accuracy would potentially be increased by increasing the number of flashes due to the speed/accuracy tradeoff. However, we have shown in our previous publications that any decrease in accuracy as a result of this reduction is minor in comparison to the increase in speed (Speier, 2015).

 In page 2 line 60-62 the authors state: « In the P300 speller, the flash duration for a single target selection and the ISI are both 62.5 milliseconds »---Do you mean that any P300 speller uses these timing ? This is not correct.

We meant that those are the parameters we used in our current study. The text of this section has been modified to make this more clear.

In the language model section the authors mention Figure 2 (line 135) however Figure 2 corresponds to the COMB pattern.

We apologize for the error. We removed several unnecessary figures from a previous draft and forgot to remove the references. This has been corrected.

In the Predictive Spelling section the authors mention Figure 1 (line 224) however Figure 1 corresponds to CBP pattern.

We apologize for the error. As mentioned in the previous response, we removed several figures from a previous draft and forgot to remove the figure references. 

This has been corrected.

Figure 3 is not mentioned into the text

A reference to figure 3 has been added to the text.

---

## [Decision Letter · Decision Letter 1]

3 Jan 2024

PONE-D-22-19793R1Comparing P300 flashing paradigms in online typing with language modelsPLOS ONE

Dear Dr. Speier,

Thank you for submitting your manuscript to PLOS ONE. After careful consideration, we feel that it has merit but does not fully meet PLOS ONE’s publication criteria as it currently stands. Therefore, we invite you to submit a revised version of the manuscript that addresses the points raised during the review process.

Please, address the issues concerning statistical analysis expressed by the reviewer #1. 

We look forward to receiving your revised manuscript.

Kind regards,

Gennady S. Cymbalyuk, Ph.D.

Academic Editor

PLOS ONE

Journal Requirements:

Additional Editor Comments:

Please, address the issue with the statistical analysis of the data as suggested by the reviewer #1. Line 244-255 and Line 256-264: Were statistical tests carried out within one flashing pattern, comparing its performance over different amounts of training data, or within one amount of training data, comparing different flashing patterns with each other? From the text, it seems that the first strategy was chosen, but to verify the claim whether the flashing pattern has impact on performance, the second strategy should be used.

Reviewers' comments:

Reviewer's Responses to Questions

**Comments to the Author**

1. If the authors have adequately addressed your comments raised in a previous round of review and you feel that this manuscript is now acceptable for publication, you may indicate that here to bypass the “Comments to the Author” section, enter your conflict of interest statement in the “Confidential to Editor” section, and submit your "Accept" recommendation.

Reviewer #1: (No Response)

Reviewer #3: All comments have been addressed

2. Is the manuscript technically sound, and do the data support the conclusions?

Reviewer #1: Partly

Reviewer #3: Yes

3. Has the statistical analysis been performed appropriately and rigorously? 

Reviewer #1: No

Reviewer #3: Yes

4. Have the authors made all data underlying the findings in their manuscript fully available?

Reviewer #1: Yes

Reviewer #3: Yes

5. Is the manuscript presented in an intelligible fashion and written in standard English?

Reviewer #1: Yes

Reviewer #3: Yes

6. Review Comments to the Author

Reviewer #1: Table 2: Replace "Reimann" with "Riemann"

Line 244-255: It is mentioned that "None of these decreases was statistically significant" Later, it is mentioned that "The CBP performance was not significantly different from the full dataset". Were statistical tests carried out within one flashing pattern, comparing its performance over different amounts of training data, or within one amount of training data, comparing different flashing patterns with each other? From the text, it seems that the first strategy was chosen, but to verify the claim whether the flashing pattern has impact on performance, the second strategy should be used.

Line 256-264: Similar remark. the text tests and discusses whether there is an impact of the classifier of performance within the different flashing patterns, not whether there are differences in performance across flahsing patterns within a given classifier. Only the latter strategy contributes to verifying whether the flashing patterns impact performance, which is, to the reviewer's understanding, what is investigated here.

Reviewer #3: In this paper authors have tested different flashing patterns with other optimization techniques on P300 speller performance. Authors did not find significant difference and conclude that optimizing flashing patterns is not improve P300 speller performance. Negative result is also important result.

7. PLOS authors have the option to publish the peer review history of their article (what does this mean?). If published, this will include your full peer review and any attached files.

Reviewer #1: No

Reviewer #3: No

---

## [Author Response · Author response to Decision Letter 1]

23 Mar 2024

Reviewer #1: Table 2: Replace "Reimann" with "Riemann"

We have corrected this typo.

Line 244-255: It is mentioned that "None of these decreases was statistically significant" Later, it is mentioned that "The CBP performance was not significantly different from the full dataset". Were statistical tests carried out within one flashing pattern, comparing its performance over different amounts of training data, or within one amount of training data, comparing different flashing patterns with each other? From the text, it seems that the first strategy was chosen, but to verify the claim whether the flashing pattern has impact on performance, the second strategy should be used.

We thank the reviewer for this clarification and agree that the analysis they propose is more appropriate. When reducing the training set, the COMB paradigm still produces the highest ITR in offline experiments. This difference remains significant in the 50% case. When reducing the training set to 30%, however, the difference between COMB and CBP is no longer significant. This analysis has been added to the results section.

Line 256-264: Similar remark. the text tests and discusses whether there is an impact of the classifier of performance within the different flashing patterns, not whether there are differences in performance across flashing patterns within a given classifier. Only the latter strategy contributes to verifying whether the flashing patterns impact performance, which is, to the reviewer's understanding, what is investigated here.

Since the primary goal of this paper was to explore the performance of the models during online experiments, we believed that the main aim when comparing classifiers offline was to justify the use of LDA in online experiments. However, we recognize the interest in investigating whether the trends in LDA extend to other classifiers. We have therefore adjusted this paragraph to include results comparing flashing paradigms. Overall, the trend of COMB having the best ITR is persistent across classifiers, although it is only significant when using LDA and RF.

---

## [Decision Letter · Decision Letter 2]

5 Apr 2024

Comparing P300 flashing paradigms in online typing with language models

PONE-D-22-19793R2

Dear Dr. Speier,

We’re pleased to inform you that your manuscript has been judged scientifically suitable for publication and will be formally accepted for publication once it meets all outstanding technical requirements.

Kind regards,

Gennady S. Cymbalyuk, Ph.D.

Academic Editor

PLOS ONE

Additional Editor Comments (optional):

Reviewers' comments:

Reviewer's Responses to Questions

**Comments to the Author**

1. If the authors have adequately addressed your comments raised in a previous round of review and you feel that this manuscript is now acceptable for publication, you may indicate that here to bypass the “Comments to the Author” section, enter your conflict of interest statement in the “Confidential to Editor” section, and submit your "Accept" recommendation.

Reviewer #1: All comments have been addressed

2. Is the manuscript technically sound, and do the data support the conclusions?

Reviewer #1: Yes

3. Has the statistical analysis been performed appropriately and rigorously? 

Reviewer #1: Yes

4. Have the authors made all data underlying the findings in their manuscript fully available?

Reviewer #1: Yes

5. Is the manuscript presented in an intelligible fashion and written in standard English?

Reviewer #1: Yes

6. Review Comments to the Author

Reviewer #1: No further comments as all my concerns were addressed. And the data will be uploaded to the OpenNeuro repository upon publication.

7. PLOS authors have the option to publish the peer review history of their article (what does this mean?). If published, this will include your full peer review and any attached files.

Reviewer #1: No

---

## [Editor Report · Acceptance letter]

25 Apr 2024

PONE-D-22-19793R2 

PLOS ONE

Dear Dr. Speier, 

I'm pleased to inform you that your manuscript has been deemed suitable for publication in PLOS ONE. Congratulations! Your manuscript is now being handed over to our production team.

Kind regards, 

on behalf of

Dr. Gennady S. Cymbalyuk 

Academic Editor

PLOS ONE